



**An Assessment of the Impact of ATMS and CrIS Data Assimilation on Precipitation**
**Prediction over the Tibetan Plateau**
Tong Xue
Key Laboratory of China Education Ministry for Meteorological Disasters, Nanjing University
of Information Science and Technology, Nanjing, China
Guangdong Ocean University, Zhanjiang, China
Jianjun Xu
Guangdong Ocean University, Zhanjiang, China,
GENRI, College of Science, George Mason University, Fairfax, Virginia, USA
Zhaoyong Guan
Nanjing University of Information Science and Technology, Nanjing, China
Long S. Chiu
AOES, College of Science, George Mason University, Fairfax, Virginia, USA,
Han-Ching Chen
Department of Atmospheric Sciences, National Taiwan University, Taipei, Taiwan
Min Shao
GENRI, College of Science, George Mason University, Fairfax, Virginia, USA
*Corresponding author contact information: Dr. JIANJUN XU,  Guangdong Ocean University,
Zhanjiang, China.  Email: jxu14@gmu.edu



## Abstract

Using the National Oceanic and Atmospheric Administration's Gridpoint Statistical Interpolation data assimilation system and the National Center for Atmospheric Research's Advanced Research Weather Research and Forecasting (WRF-ARW) regional model, the impact of assimilating advanced technology microwave sounder (ATMS) and cross-track infrared sounder (CrIS) satellite data on precipitation prediction over the Tibetan Plateau in July 2015 was evaluated. Four experiments were designed: a control experiment and three data assimilation experiments with different data sets injected: conventional data only, a combination of conventional and ATMS satellite data, and a combination of conventional and CrIS satellite data. The results showed that the monthly mean of precipitation is shifted northward in the simulations and shows an orographic bias described as an overestimation in the upwind of the mountains and an underestimation in the south of the rainbelt. The rain shadow mainly influenced prediction of the quantity of precipitation, although the main rainfall pattern was well simulated. For the first 24-hourand last 24-hour accumulated daily precipitation, the model generally overestimated the amount of precipitation, but it was underestimated in the heavy rainfall periods of 3-6, 13-16, and 22-25 July. The observed water vapor conveyance from the southeastern Tibetan Plateau was larger than in the model simulations, which induced inaccuracies in the forecast of heavy rain on 3–6 July. The data assimilation experiments, particularly the ATMS assimilation, were closer to the observations for the heavy rainfall process than the control. Overall, the satellite data assimilation can enhance the



WRF-ARW model's ability to predict the spatial and temporal pattern of precipitation in July 2015
although the model capability exists a significant limitation in the complex terrain area.
**Key words:** Radiance data assimilation, GSI, Tibetan Plateau, Weather forecast accuracy



## 1. Introduction

The Tibetan Plateau (TP) is the highest and largest plateau in the world. It is located in the central Eurasian continent and stands in the middle troposphere, covering an area of approximately 2.5 million $km^2$. The TP has a variety of topographical features of a large terrain gradient and its steep mountains are aligned with an east-to-west arrangement. The dramatic modification caused by the rugged terrain influence the local atmospheric circulation and cause strong local convection to arise, easily inducing severe weather such as heavy rainfall, windstorms, hailstorms, and so on (Massacand et al., 1998; Gao et al., 2015). Precipitation is one of the key variables for understanding the hydrological cycle on the TP and has profound effects on the regional and global circulation that affect millions of people in the adjacent areas (Ye and Gao, 1979; Chen et al., 1985; Chambon et al., 2014; Li et al., 2014). Therefore, making accurate and long-lead weather forecasts at high temporal and spatial resolution for the TP not only has scientific significance but also addresses the urgent need for disaster prevention. However, due to the variable weather conditions and complex terrain orography, the TP remains a sparsely populated region with few conventional observation data sources, and the limited available meteorological data leads to great uncertainties in the regional weather forecasts. The continuous development of numerical weather prediction (NWP) models, such as the National Center for Atmospheric Research (NCAR)'s Advanced Research Weather and Research Forecasting (WRF-ARW) model, offer opportunities to improve regional weather forecasts in data-sparse regions. NWP models can be initialized with and laterally assimilate observation data, which is beneficial for better describing atmospheric conditions, thus



keeping model results close to observations (Maussion et al., 2011).
Satellite radiance data are one of the most important observation data sources and can be
directly assimilated into data assimilation models. Compared with conventional observation data,
geostationary satellite data have continuous spatial and temporal coverage and polar orbiting
satellites circle the earth twice a day to provide global observations of multiple meteorological
variables, such as temperature, pressure, moisture, and so on. Moreover, many studies have
suggested that the assimilation of satellite radiance data can substantially improve weather
forecasts (Eyre, 1992; Derber and Wu, 1998; Xu et al., 2009). For longer-range prediction, satellite
data are even more crucial than conventional observations (Zapotocny et al., 2008), Past studies
have also indicated that the effect of assimilation of both observations and satellite products was
better than only satellite data assimilation (Liu et al., 2013).However, the performance of satellite
radiance assimilation in limited-area modeling systems using variational DA method is still
controversial (Zou et al., 2013; Newman et al., 2015).Schwartz et al. (2012) was the first to
assimilate microwave radiances with the region lacking observation stations using ensemble
Kalman filter (ENKF) and the results showed that assimilating microwave radiances overall make
better forecasts of Typhoon Morakot (2009). The negative influence has also appeared and it is
mainly contributed to various of factors such as the influence of lateral boundary conditions within
the regional domain (Warner et al., 1997) and non-uniform satellite coverage (Kazumori et al.,

2013). .

The advanced technology microwave sounder (ATMS) and cross-track infrared sounder (CrIS)



are two instruments with high resolution onboard the Suomi National Polar-orbiting Partnership
spacecraft , a polar-orbiting satellite launched in 2011 with the aim to provide real-time sensor
data for critical weather and climate measurements. The ATMS, a cross-track microwave scanner
with 22 channels, combines most of the channels of the preceding advanced microwave sounding
unit (AMSU-A) and microwave humidity sounder (MHS) to provide sounding profiles of
atmospheric moisture and temperature. The CrIS is a Fourier transform spectrometer with 1305
spectral channels inherited from the high-resolution infrared radiation sounder (HIRS) to produce
temperature, pressure, and moisture profiles. A previous study assimilated ATMS data in the
European Centre for Medium-Range Weather Forecasts system and the results showed that the
instrument has better performance than AMSU-A and MHS in the longer range over the Northern
Hemisphere (Bormann et al., 2013). Nevertheless, satellite data assimilation into NWP models
over the TP presents special challenges, because the limited model capability for assimilating
radiance data over complex terrain with heterogeneous characteristics is still not clearly recognized.
Furthermore, whether the new generation of satellite observations, such as ATMS and CrIS, can
compensate for the shortage of data over the TP and effectively enhance the accuracy of forecasts
remains unknown.
In this paper, we make an assessment of the impact of assimilating ATMS and CrIS radiance
data for East Asia on precipitation prediction over the TP and compare the effects of different
satellite data sets injected.
**2. Data and Models**


*2.1 Data*
*2.1.1 Background data*
The National Centers for Environmental Prediction (NCEP) global forecast system (GFS)
forecast data, which has a horizontal resolution of $0.5° × 0.5°$ with a 6-hour interval, were used as
the boundary and initial conditions for the control (CTRL) experiment, while the background fields
of data assimilation experiments (DA) take advantages of the forecast product at 0600 UTC made
by CTRL. The GFS data is publicly available from https://www.ncdc.noaa.gov/data-access/model-
data/model-datasets/global-forcast-system-gfs.

*2.1.2 Observation data*
Observational precipitation data from the National Meteorological Information Center (NMIC)
of the China Meteorological Administration (CMA) was used as the truth data for comparison with
the model results. The $0.1° × 0.1°$ high-resolution gridded hourly China Merged Precipitation
Analysis (CMPA) data gauge, which combines the CMA's rain gauge hourly data provided by
more than 30,000 automatic weather stations with the National Oceanic and Atmospheric
Administration (NOAA) Climate Prediction Center's Morphing (CMORPH) precipitation product
(Xie & Xiong, 2011; Pan et al., 2012; Shen et al., 2014), was used for verification to evaluate the
model simulation results. Previous study (Guo et al., 2014) have compared the CMORPH data
with other data sets in the TP area. Considering the topographically complex terrain characterizing
the TP, satellite precipitation data with very high spatial resolution is especially needed. Of the



Several merged satellite precipitation products (i.e.TMPA, PERSIANN, and GSMaP), the
CMORPH product has the highest resolution (8 km) and so it is the most suitable product to use
in studying precipitation patterns over the TP (Guo et al.,2014). The spatiotemporal resolution of
CMORPH products is also considerably higher than that of TRMM 3B42 products. Past study has
also made a conclusion that the high resolution CMORPH data can depict finer regional details,
such as a less coherent phase pattern over the TP and better capture the features of the diurnal cycle
of summer precipitation compared with TRMM 3B42(Zhang et al.,2015).

NCEP Final Analysis (FNL) data was used through dynamic downscaling as observed

moisture to illustrate the transportation of water vapor in East Asia.

*2.1.3 Assimilation data*

The conventional data which is from the GDAS-prepared BUFR files (gdas1.tCCz.prepbufr.nr)

is composed of a global set of surface and upper air reports operationally collected by the National
Centers for Environmental Prediction (NCEP). It includes radiosondes, surface ship and buoy
observations, surface observations over land, pibal winds and aircraft reports from the Global
Telecommunications System (GTS), profiler and US radar derived winds, SSM/I oceanic winds
and TCW retrievals, and satellite wind data from the National Environmental Satellite Data and
Information Service (NESDIS). The reports can include pressure, geopotential height, temperature,
dew point temperature, wind direction and speed. (National Centers for Environmental
Prediction/National Weather Service/NOAA/U.S. Department of Commerce. 2008, updated



daily. *NCEP ADP Global Upper Air and Surface Weather Observations (PREPBUFR format),*
*May 1997 - Continuing.* )
ATMS and CrIS satellite radiance data are also from the Global Data Assimilation System
(GDAS) which is in the BUFR format. All of this can be downloaded from
https://www.ncdc.noaa.gov/data-access/model-data/model-datasets/global-data-assimilation-
system-gdas.

*2.2 Models*
*2.2.1 WRF-ARW regional model*
NCAR's WRF-ARW regional model associated with the Gridpoint Statistical Interpolation
(GSI) data assimilation system was used in this study. WRF-ARW is a fully compressible
nonhydrostatic, primitive-equation, mesoscale meteorological model. As shown in Figure 1a, the
model domains are two-way nested with 12 km (580 × 422) and 4 km (817 × 574) horizontal
spacing. There are 51 vertical levels with a model top of 10 hPa. Figure 1 shows that D01 is set to
cover most of East Asia and the subdomain (D02) inside corresponds to the Tibetan Plateau, which
has a mountain–valley structure.
The physical parameterizations chosen for the forecast model in this study followed previous
studies of the area (He et al., 2012; Xu et al., 2012; Zhu et al., 2014). These included the WRF-
ARW Single-Moment 6-class (WSM-6) microphysics scheme, the Kain-Fritsh (KF) cumulus
parameterization, the Rapid Radiative Transfer Model (RRTMG) longwave and shortwave



radiation, the Yonsei University scheme (YSU) and the Noah Land Surface Model for the planetary
boundary layer scheme.

*2.2.2 GSI system and Community Radiative Transfer Model*

In this study, we chose to use the GSI 3D-Var system, which is a data assimilation system that

was initially developed as the next-generation analysis system based on the operational Spectral
Statistical Interpolation (SSI) at NCEP (Derber and Wu, 1998). .
The development of fast radiative transfer models has allowed for the direct assimilation of
satellite infrared and microwave radiances in NWP systems (Saunders et al., 1999; Gauthier et al.,
2007; Zou et al., 2011). The Community Radiative Transfer Model (CRTM) developed by the
United States Joint Center for Satellite Data Assimilation (JCSDA) has been incorporated into the
NCEP GSI system to rapidly calculate satellite radiances (Han, 2006; Weng, 2009). In this study,
the ATMS and CrIS satellite radiance data can be read in GSI via CRTM 2.1.3. It is worth noticing
that the CrIS scans a 2200km swath width (+/- 50 degrees), with 30 Earth-scene views. Each field
consists of 9 fields of view, arrayed as 3x3 array of 14km diameter spots (nadir spatial resolution).
( https://jointmission.gsfc. nasa.gov/cris.html). The ATMS scans a 2300km swath width with 96
Earth-scene views. The 1-2 channel of the spatial resolution of ATMS at nadir is 75km; 3-6
channel is 32km; 17-22 channel is 16km (Dong et al., 2014).
*2.3 Radiance data quality control*



As the quality of the observational data is easily affected by the observation instruments,
station positions, or human factors, carrying out quality control before data application is necessary
(Hubbard and You, 2005). Before data assimilation, a multiple-step quality control procedure was
applied to the satellite radiance data in the GSI system and preprocessed by NOAA's Satellite and
Information Service (NESDIS). Besides data thinning, it can be summarized to several quality
control (QC) categories in GSI to either toss the questionable observations or inflate the low
confidence observations. QC1: the observation-minus-model index based on liquid water path and
brightness temperature is calculated to check the cloud affected profile; QC2: toss the channel of
inaccurate emissivity/surface temperature estimate over sea; QC3: inflate observation error over
high terrain; QC4: retrieved the profiles which meet criterion in QC1 and QC2. The observational
number of ATMS data ranging from 53042 to 68618 in contrast to the number of CrIS data ranging
from 2694048 to 3454542 are read in DA system. After the data had passed rigorous quality
assessment and quality control processes, the results showed that about 23.2%-26.4%, and 1.3%
and 1.6% of "good" observations related to ATMS and CrIS read data separately were retained
after quality control (Fig. 2). This difference can be explained that CrIS has 1305 channel satellite
radiance data, but the number of assimilated channels are significantly reduced (Table 2),  the
selection of redundant channel leads to some part of observation radiance data comes from the
similar altitude and contains large amount of repeated information. Therefore, larger percentage of
CrIS satellite radiance data than ATMS is tossed through QC steps. Figure 1(b) shows the
distribution of the conventional data at 06:00 UTC on 1 July 2015, where observational data are



rare in the TP. Figure 1c and 1d displays the distribution of satellite data after quality control,
where there is almost complete spatial coverage in East Asia including the TP.

**3. Method and experimental design**
*3.1 Method*

A basic two-by two contingency table (Table 1) was generated to calculate the Bias Score

(BIAS), Fraction skill Score (FSS), Equitable Threat Score (ETS), Probability of False Detection
(POFD), Probability of Detection (POD), and False Alarm ratio (FAR).

The BIAS, which measures the ratio of the frequency of forecast events to the frequency of

observed events, is defined as:
$\text{BIAS} = \frac{\text{Hits+False alarms}}{\text{Hits+Misses}}$                    (1)
The FSS introduced by Roberts and Lean (2008) is a neighborhood verification method. The FSS
is defined as:
$\text{FSS} = 1 - \frac{FBS}{FBS_{ref}}$                    (2)
Fractions Brier Score (FBS) is presented as
$\text{FBS} = \frac{1}{N}\sum_{i=1}^{N}[F_o - F_f]^2$                    (3)
Where $N$ is the number of all grid points in the domain. $F_o$ and $F_f$ are the observation and forecast
fractions of the sliding window at each grid point. The sliding window in this study is 100km (25
grid points). The reference Fractions Brier Score ($FBS_{ref}$) represent a largest possible FBS and is
given as :



$\quad FBS_{ref} = \frac{1}{N}\left[\sum_{i=1}^{N} F_o{}^2 + \sum_{i=1}^{N} F_f{}^2\right]$ (4)
The ETS computes the fraction of observed events that were correctly predicted:
$\quad \text{ETS} = \frac{\text{Hits}-R}{\text{Hits}+\text{False alarms}+\text{Misses}-R}$ (5)
where R is the random forecast coefficient, given by:
$\quad R = \frac{(\text{Hits}+\text{False alarms})(\text{Hits}+\text{Misses})}{(\text{Hits}+\text{False alarms}+\text{Misses}+\text{Correct rejections})}$ (6)
The POFD measures discrimination:
$\quad \text{POFD} = \frac{\text{False alarms}}{\text{False alarms}+\text{Correct rejections}}$ (7)
Similar to the POFD, the POD shows the hits out of total observed events:
$\quad \text{POD} = \frac{\text{Hits}}{\text{Hits}+\text{Misses}}$ (8)
The FAR indicates the fraction of the predicted events that did not occur:
$\quad \text{FAR} = \frac{\text{False alarms}}{\text{Hits}+\text{False alarms}}$ (9)
To compare the model simulation data with the observation data, the 4-km model grid was
interpolated to observation data with 0.1°×0.1° degree grid based on linear interpolation method.

*3.2 Experimental design*
Four one-month-long experiments were conducted (Fig. 3). The CTRL experiment was carried
out first with an initial time of 00:00 UTC and made 54 h forecasts. The data assimilation was
applied on the D01 region of the output from CTRL at 06:00 UTC. The DA experiments made use
of the assimilated D01 and the D02 from the CTRL at 06:00 UTC as the initial condition and made
a 48 h forecast for each day. Three DA experiments were performed with a time window of 3 hours:



(1) a conventional run (CONV) assimilating the conventional observation data only; (2) an ATMS
radiance run (ATMS) adding the ATMS satellite radiance data to the CONV; and (3) a CrIS
radiance run (CRIS) adding the CrIS satellite radiance data to the CONV.

The accumulated precipitation integrated from 06 to 30 h and 30 to 54 h are defined as the

first twenty-four-hour accumulated (F24H) precipitation and last twenty-four-hour accumulated
(L24H) precipitation, respectively.

**4. Results**
*4.1 Impact of DA on the spatial fields of precipitation forecast*

Figure 4 shows the spatial pattern of the monthly mean of 24-hour accumulated precipitation

in July 2015. Monthly mean precipitation exhibits a decreasing south-to-north gradient. The
predicted precipitation in the central and northern parts of the TP, Qaidam Basin (90°-99°E, 35°-
39°N), Tarim Basin (75°-90°E, 37°-42°N), and Junggar Basin (80°-90°E, 45°-48°N) was too small
to be measured (Fig. 4a, c). It was found that F24H precipitation ranged from 6.0 to 30.4 mm,
while the L24H forecasts ranged from 6.0 to 29.5 mm per month. The rain shadow along the
Himalayas (73°-95°E, 27°-35°N) was found in the spatial distribution of precipitation. The CTRL
(Fig. 4b, d) mostly simulated the monthly mean rainbelt distributed along the southern and
southwestern margin of the plateau, between the Himalayas in the west and the Hengduan
Mountains (95°-103°E, 24°-32°N) in the east. The difference between the model simulations and
observations (Fig. 5) indicated that the CTRL simulation tends to overestimate precipitation,



especially in the southern and southwestern margin along the rainbelt where the altitude changes
from 500 to 3000 m. The results suggested that the WRF-ARW model has limitations in simulating
the precipitation in mountainous areas, which is similar to the conclusion of previous studies (He
et al., 2012; Xu et al., 2012). Furthermore, we found that the precipitation is overestimated (colored
red) in the upwind of the mountains along the southwestern margin. In contrast, the precipitation
is underestimated in the south of the rainbelt, leading to a north–south dipole structure. This pattern
results in a northward migration of the rainbelt in the simulations. The three DA experiments
indicated that the assimilation of satellite radiance data can not calibrate the rain shadow effect
and all experiments showed consistently gross overestimation patterns, varying from 8 to 10 mm
about the monthly mean precipitation. The overall bias statistic in D02 is 0.97 mm (0.86 mm), 0.52
mm (0.70 mm), 1.08 mm (0.97 mm), and 0.98 mm (0.76 mm) CTRL, CONV, ATMS and CRIS
respectively. This may be attributed to the physical package of WRF-ARW having an inadequate
description of snow cover over the plateau surface making the error of margin more prominent
(Marteau et al. 2015).

Figure 6 shows the spatial patterns according to the contingency table (Table 1) and the scatter

plots, in which monthly mean 24 h rainfall over the 6 mm threshold is defined as an "event".
Rainfall events occur over most of the TP area, including the northern Gangetic Plain (80°-90°E,
24°-28°N) where the elevation is lower than 3000 m, and can be well predicted with ~8–10% hits
(A) and ~76–79% correct rejections (D) in the majority of the region. The false alarms (B) were
spread mainly in the east of the TP, where the Bayan Har (95°E, 35°N) and Hengduan mountains



are located, accounting for ~7–10%, while the misses (C) were distributed in the western plain
exterior of the TP and accounted for ~5–6%. It's also evident to see the dipole pattern in the
distribution of the hits and misses similar to the Figure 5. Among the four linear regression lines
(bold grey lines), ATMS looks a little better than the other three experiments but has more extreme-
precipitation event forecasts than the others, followed by the CTRL and CRIS, while CONV has
the lowest simulation precision. This implies that the rain shadow mainly influences prediction of
the quantity of precipitation, but rainfall events are still well predicted. Furthermore, the
comparisons indicated that the WRF-ARW model has promising potential, as the false alarms were
primarily located in the east of the TP in contrast to the misses in the west.

Figure 7 shows the monthly and domain average validation statistics in the TP. The differences

between the four experiments for the F24H forecasts are larger than for the L24H forecasts. The
ETS, FSS, and POD values all decline as the threshold increases; a higher value for these three
skill scores indicates a better performance of the experiments. ATMS showed the highest FSS (Fig.
7b), ETS (Fig. 7c) and POD (Fig. 7d). CONV performed similar to the CTRL in ETS and FSS,
and CRIS performed the worst. However, according to the BIAS, CONV is mostly approximately
1, which indicates the best overall relative frequencies compared with the other experiments.
Through the 1–5 mm threshold, CRIS performs the largest overforecast (BIAS > 1), but it evolves
to have a better performance than ATMS and CTRL through the 5–10 mm threshold. FAR and
POFD results indicate that CONV performs best (0 is perfect), followed by ATMS and then CTRL
and CRIS. However, POD results manifest that ATMS performs best (1 is perfect) and CONV is



worst. The different methods of forecast verification may depend on the purpose of the verification,
and the results we evaluated by different methods can explain the different question we want to
answer. Overall, the results reflect that DA has a positive effect on reproducing the monthly mean
precipitation in the TP compared with the CTRL to varying degrees.

*4.2 Impact of DA on the temporal distribution of precipitation forecast*
Another measure of performance is to examine how the daily precipitation is temporally
distributed (Fig. 8). It can be seen in the time series that there are four observed heavy rainfall
events(3.0 mm/day) during the periods of 3–6, 8-10,13–16 and 22–25 July (Fig. 8a). In general,
the 24 h amount of precipitation is overestimated in all three DA experiments by 20%, 40%, and
37% for CONV, ATMS, and CRIS, respectively. In contrast, of the 4 heavy rainfall periods, 3
events including 3–6, 13–16 and 22–25 July are underestimated. The L24H forecasts showed a
similar pattern, except that there were much smaller differences between the three DA experiments
compared with the F24H forecasts. The F24H forecasts appear the one-day time lag effect
compared with L24H. Because the F24H forecasts calculate the cumulative precipitation of the
first 6-30 hour while the L24H forecasts represent the 30-54 hour cumulative precipitation
forecasts. When all the overestimation events are considered, the CONV experiment captured the
accumulated amount of precipitation much more accurately than the other DA experiments and the
ATMS performed the worst. As mentioned above, the 24 h precipitation maximums surpassing 20



mm are spread in the main precipitation region, showing that the prominent geographical
dependence of rainfall coincides with the threshold of heavy rainfall defined for TP areas.
Although previous studies and our results show an obvious trend of overestimating rainfall
in the TP, there appears to be underestimated during heavy rainfall events (Fig. 8). To determine
the forecast capabilities of the model in the heavy rainfall periods, we focused on one heavy rainfall
period of 3-6 July.
Figure 9 shows the rainfall intensities (bars) calculated for every 3 h amount of precipitation.
The cumulative precipitation (curves) is defined as the precipitation accumulated for each 3 h
starting at 06:00 UTC during 3–6 July. From the perspective of observations, this rainfall event
can be divided into three periods, of which the 3 July is ahead of the heavy rainfall with less than
0.45 mm per 3 h, followed by the rainfall around 03:00 UTC on 4 July to 03:00 UTC on 5 July,
with the first peak at 21:00 UTC on 4 July of more than 0.65 mm per 3 h. The third phase started
at 03:00 UTC on 5 July and ended at 00:00 UTC on 6 July with a second rainfall pulse around
21:00 UTC on 5 July exceeding 0.60 mm per 3 h and then weakening. It is evident that this rainfall
event had a significant diurnal harmonic and the maximum precipitation always occurred at 18:00–
21:00 UTC (00:00–03:00 LST). This diurnal variation was remarkable, especially when the heavy
rainfall occurred, which was equivalent to evening local solar time (LST). However, the simulated
maximum always occurred at 06:00–09:00 UTC (12:00–15:00 LST), earlier than the observations,
and can probably be attributed to the limit of complicated topography. In this case, simulated
rainfall intensity was much lower than the observations during 09:00 UTC on 4 July to 00:00 UTC



on 5 July and 12:00 UTC on 5 July to 21:00 UTC on 5 July when the rainfall occurred. That is, the
model can not promptly quantitatively predict the sudden occurrence of this event. Moreover, the
cumulative curves of the model show an overestimation on 3 and 5 July compared with
observations; in particular, the cumulative curves of the CTRL are far away from the measured
values due to an inaccurate initial field. It can be concluded that the DA experimentsdata are closer
to the observations during the heavy rainfall period compared with the CTRL experiment.

*4.3 Impact of DA on circulation and water vapor supply*
According to the above-mentioned analysis, it is evident that DA improves forecasts during
the heavy rainfall period, but the results are not universal when different data sets are injected. As
is well known, adequate water vapor transport is one of the preconditions for precipitation
formation. In this section, we discuss the water vapor supply in the 3–6 July case study, with the
aim of determining the reason for the different influences exerted by different experimental
schemes. Figure 10 shows the F24H forecasts of precipitation quantity (shadings) and water vapor
flux (vectors) during 3–6 July. According to observations, warm and humid water vapor is
transferred from the Bay of Bengal eastward by the southwest monsoon. The TP blocks the
westward transport of humid and warm air, and this rainfall event start developing in the southeast
of the TP on 3 July and then the rainbelt runs southeast to southwest and develops along the
Himalayas on 4–5 July. Comparing the observations (Fig. 10a–c) with model results (Fig. 10d–f),
the simulated precipitation is considerably larger than the observed on 3 July before the heavy



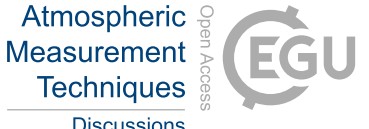

rainfall occurs, but as time goes on this condition reverses. For the difference value distribution
(Fig. 10g–i) of the CTRL minus observations, the main water vapor flux divergence differences
(shadings) are negative in the rainy region on 3 July, which indicates that the water vapor
convergence is stronger than observed, inducing the overestimation. However, when the rainfall
event occurs on 4–5 July, this condition is opposite. The water vapor differences (vectors) also
suggest that the observed water vapor conveyance from the southeastern of the TP is larger than
the model simulation, which induces inaccuracies in the forecast of the heavy rain. Therefore,
analysis of moisture is useful for improving the heavy rainfall forecasting skill.

To further discuss the effect of DA on this rainfall event, the differences between the simulated

F24H precipitation and the observed distribution and the ETS skill scores (Fig. 11) were considered.
From the spatial distribution, all the experiments (Fig. 11a, d, g, j) overestimated the precipitation
quantity, especially the CTRL, before the heavy rainfall and the FSS skill scores all ranged from
0.46 to 0.49 with little differences (left in Fig. 11m). When the heavy rainfall event occurred on 4
July, the observed rainbelt moved southwest (Fig. 11b, e, h, k), while the simulated rainbelt was
motionless, leading to an underestimation in the southwest. The FSS scores for ATMS, CONV and
CTRL ranged from 0.42 to 0.48  (middle in Fig. 11m), but CRIS only scored 0.36. As the water
vapor conveyance directly contributes to the westward movement of the rainbelt and the intensity
of this precipitation event on 5 July, the precipitation experiments all underestimated the amount
of precipitation, and CRIS performed particularly badly (Fig. 10c, f, i, l). However, ATMS had a
substantially high FSS scores (0.47) (right in Fig. 11m), followed by CRIS(0.45) and CONV(0.43)



while CTRL only scored 0.35. This phenomenon indicates that DA can indeed improve the heavy
rainfall forecast. From the above analysis of Figure 9 and 10, it is clear that before the heavy
rainfall, DA can improve the simulation of precipitation spatially. As time passes and the heavy
rainfall develops, DA, especially the ATMS assimilation, can enhance model prediction abilities
both spatially and temporally in comparison with the CTRL experiment.

**5. Summary and discussion**

In this study, we used diagnostic methods to analyze the impact of DA on the monthly

precipitation distribution over the TP and then focused on one heavy rainfall case study that
occurred from 3 to 6 July 2015. The DA and NWP were performed for July 2015 to make the
weather forecasts. The spatial distribution of monthly mean precipitation showed an evident rain
shadow effect along the Himalayas and that the precipitation decreased northward in the TP.
However, the simulated precipitation belt was shifted northward compared with the observed
rainbelt and showed an orographic bias described as an overestimation in the upwind of the
mountains and an underestimation in the south of the rainbelt. Assimilation of satellite radiance
also can not calibrate the rain shadow effect and all experiments showed consistently gross
overestimation patterns. Furthermore, it seems that the rain shadow mainly influences prediction
of the quantity of precipitation, but the main rainfall pattern can be well predicted. Comparisons
indicate that the WRF-ARW model has promising potential, in that the false alarms are primarily
predicted in the east of the TP in contrast to the misses in the west. The DA validation statistics



also suggest that DA has a positive effect on monthly mean precipitation prediction in the TP
compared with the CTRL to varying degrees. For the time series of monthly precipitation, F24H
and L24H precipitation chiefly overestimate the amount of precipitation, which is in agreement
with previous studies, , but the amount of 24 h precipitation in the three heavy rainfall periods of
3–6, 13–16, and 22–25July is underestimated.

To further study the underestimations in the heavy rainfall events and the performance of the

WRF-ARW model and GSI DA impact, we selected a case study from 3 to 6 July. It is evident that
this rainfall event had a significant diurnal harmonic and the maximum precipitation always
occurred at 18:00–21:00 UTC (00:00–03:00 LST). This diurnal variation was remarkable,
especially when the heavy rainfall occurred. Although the model can not promptly quantitatively
predict the sudden occurrence of this rainfall event, the DA, especially the ATMS simulation are
closer to the observations for the heavy rainfall event compared with CTRL experiments. Overall,
before the heavy rainfall, DA improved the precipitation prediction spatially. As time passed and
the rainbelt moved and rainfall developed, DA enhanced the model prediction abilities both
spatially and temporally. It should be mentioned that the high altitude and complex topography of
the TP and its blocking effect on moisture transfer coming from Indian Ocean by the southwest
monsoon obviously influences the rainfall forecast. As precipitation biases indicate some extent of
spatial coherence and temporal recurrence, it is possible to provide an adapted correction method
to enhance the model precipitation prediction capabilities.



It is conspicuous that the ATMS showed better performance than CTRL, CONV, and CRIS in
the case study. Past studies have indicated that the effect of assimilation of both observations and
satellite products is better than assimilation of satellite data only, which may account for the ATMS
performing better than CONV. ATMS also performed better than CRIS. As clouds are opaque in
the infrared wave band of the spectrum and largely transparent in the microwave band, microwave
instruments are thought to perform better than infrared instruments on cloudy and rainy days,
which may explain the better performance of ATMS compared with CRIS.
In this study, we investigated the monthly precipitation distribution and a selected heavy
rainfall case in the TP using the WRF-ARW mesoscale model and the GSI data assimilation system.
Moisture and dynamic conditions were analyzed in the case study; however, thermal conditions
are also one of the direct factors leading to rainfall that need to be investigated in the future.
Furthermore, although the CrIS were assimilated large amount of satellite radiance pixels, the
general DA effect is relatively worse compared with the other three experiments. CrIS has 1305
spectral channels, some of which are redundant as they include many satellite radiance
observations from similar altitudes and contain much repeated information, which may lead to the
poor DA impact. It should take the priority to select physical sensitivity and the high vertical
resolution channels. On the other hand, the high altitude and complicated dynamic, thermal
conditions increase the difficulty of selecting channels. Therefore, only by carrying out further
research on bias correction, quality control, and channel selection can satellite radiance data play
an efficient role in TP weather forecasting.



In addition, model resolution and parameterized scheme selection are also key factors
affecting forecast quality. In this study, the parameterized schemes we chose have been applied in
previous studies of the TP. It would be worthwhile to make a comparative analysis of different
parameterized schemes with higher model resolution in the future. Furthermore, it should be noted
that due to the heavy calculation burden, this study made use of 3D-Var as the assimilation method.
Other advanced assimilation techniques, such as 4D-Var, Hybrid, and EnKF, also need to be tested.

*Acknowledgements*
The WRF-ARW model was obtained from NCAR, the GSI data assimilation system was obtained
from JCSDA, and the satellite datasets were provided by NOAA/NESDIS/STAR.  The authors are
very grateful to these agencies for the model and providing data. This work was jointly supported
by the Special Fund for Public Welfare of China Meteorological Administration
(GYHY201406024) and the National Natural Science Foundation of China (91437104, 41330425,
41130960). The corresponding author was supported by the Guangdong Ocean University
Research Funding of Air–Sea Interaction and Data Assimilation (300702/E16188). The first author
was a visiting scholar at GMU/AOES during this study and acknowledges helpful discussions
with fellow members of GMU/AOES.






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



**Table 1.** Contingency table.

| Forecast | Observed | |
|---|---|---|
| | Yes | No |
| Yes | Hits | False alarms |
| No | Misses | Correct rejections |



**Table 2.** The channels for ATMS and CrIS data have been selected in the data assimilation

| Sensor | Channels |
|---|---|
| ATMS | 1-14, 16-22 |
| CrIS | 37, 49, 51, 53, 59, 61, 63, 65, 67, 69, 71, 73, 75, 77, 79, 80, 81, 83, 85, 87, 89, 93, 95, 96, 99, 101, 102, 104, 106, 107, 116, 120,123, 124,,125, 126, 130, 132, 133, 136, 137, 138, 142,143, 144, 145, 147, 148, 150, 151, 153, 154, 155, 157-168, 170, 171, 173, 175, 198, 211, 224, 279, 342, 392, 404, 427, 464, 482, 501, 529 |












**Figure captions**

**Figure 1.** (a) Simulation domains. (b)–(d) Distribution of (b) conventional data observations, (c) scan coverage of ATMS data after data assimilation, and (d) scan coverage of CrIS data after data assimilation at 06:00 UTC on 1 July 2015.

**Figure 2.** Total number of radiance observations kept and used in the forecast experiments as a function of date for the (a) ATMS and (b) CrIS data

**Figure 3.** Experiments design, CTRL: control experiment without data assimilation; CONV: data assimilation with conventional data only; ATMS: data assimilation with conventional +ATMS data; CRIS: data assimilation with conventional + CrIS data

**Figure 4.** Spatial pattern of the monthly mean precipitation (unit: mm) in July 2015. (a),(b) F24Hforecast and (c),(d) L24Hforecast. Contours are altitude (unit: m).

**Figure 5.** Difference value distribution of monthly mean precipitation (unit: mm) during July for data assimilation minus observation experiments. (a), (e) CTRL minus OBS; (b), (f) CONV minus OBS; (c), (g) ATMS minus OBS for (a)–(d) F24Hforecast and (e)–(h) L24Hforecast. Contours are altitude (unit: m).

**Figure 6.** Spatial patterns of (a)–(d) the contingency table and (e)–(h) the scatter plots (monthly mean 24 h rainfall over 6 mm threshold is defined as an "event"). The solid grey line indicate the regression line of A. Contours are altitude (unit: m).

**Figure 7.** Monthly and domain average validation statistics. (a)–(f) F24Hforecast and (g)–(l) L24Hforecast.




**Figure 8.** Daily precipitation distribution (unit: mm).

**Figure 9.** Rainfall intensities (bars) calculated for every 3 h amount of precipitation. The cumulative precipitation (curves) is defined as the precipitation accumulated for each 3 h starting at 06:00 UTC during 3–6 July. The unit is mm.

**Figure 10.** (a)–(f) 24 h forecasts of precipitation quantity (shadings) and water vapor flux (vectors) during 3–6 July for (a)–(c) OBS and (d)–(f) CTRL. (g)–(i) Differences in water vapor flux (vectors) and water vapor divergence (shadings) between CTRL and OBS. The unit of precipitation quantity is mm. The units for water vapor flux and divergence is kg/(m*s) and kg/(m$^2$*s), respectively.

**Figure 11.** Differences between (a)–(l) the simulated F24H precipitation and the observed distribution and (m) the FSS skill scores with 8 mm threshold during 3–6 July. The unit of differences is mm.



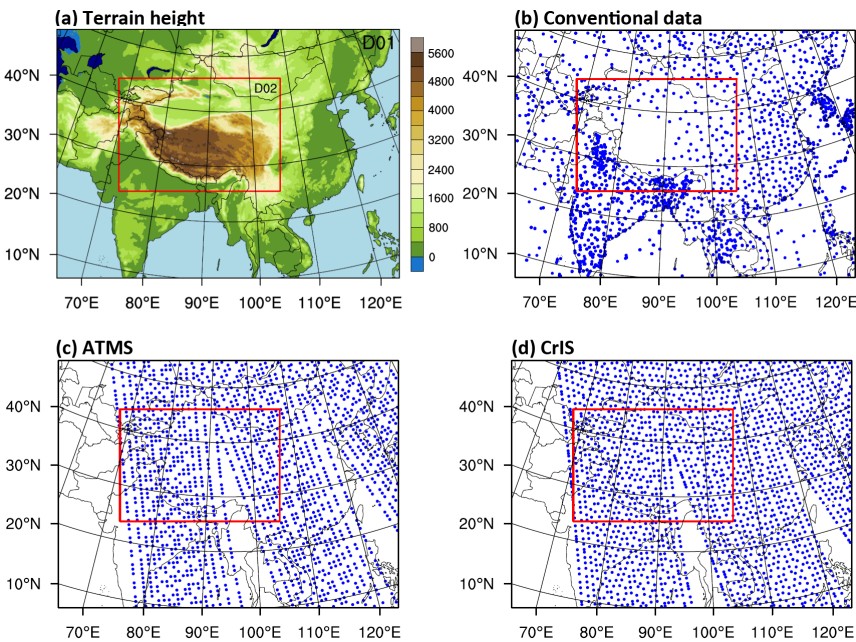

**Figure 1.** (a) Simulation domains. (b)–(d) Distribution of (b) conventional data observations, (c) scan coverage of ATMS data after data assimilation, and (d) scan coverage of CrIS data after data assimilation at 06:00 UTC on 1 July 2015.




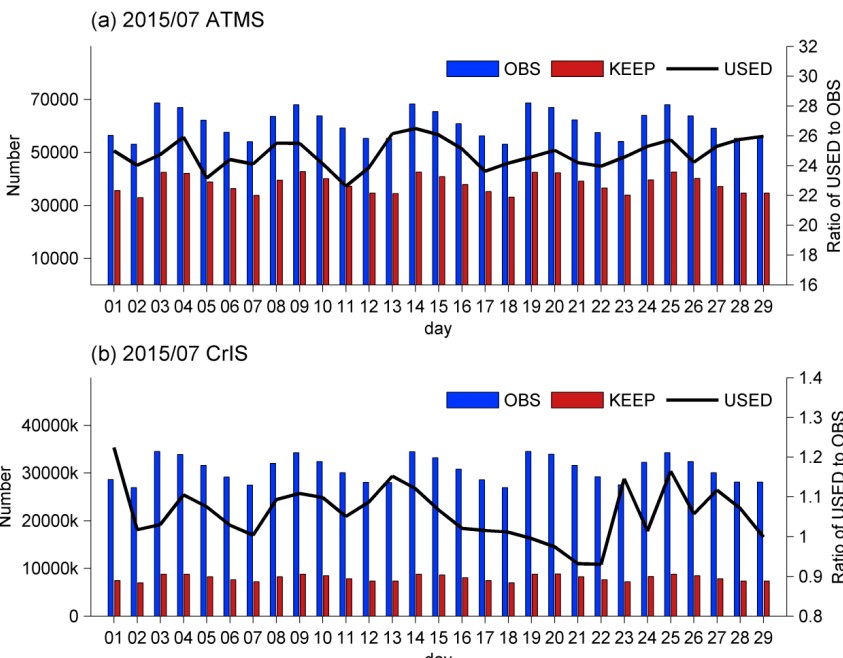


**Figure 2.** Total number of radiance observations kept and used in the forecast experiments as a function of date for the (a) ATMS and (b) CrIS

data.





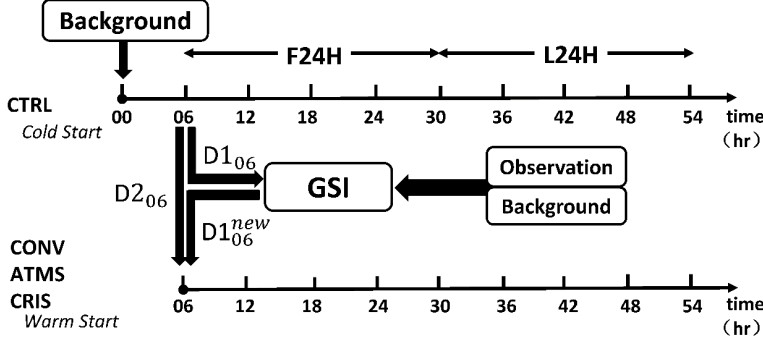

| EXP | Description | Initial time |
|------|----------------------------|-------------------------------|
| CTRL | No assimilation | 00:00 UTC from 1 Jul to 31 Jul |
| CONV | Conventional data only | 06:00 UTC from 1 Jul to 31 Jul |
| ATMS | Conventional + ATMS data | 06:00 UTC from 1 Jul to 31 Jul |
| CRIS | Conventional + CrIS data | 06:00 UTC from 1 Jul to 31 Jul |


**Figure 3.** Experiments design, CTRL: control experiment without data assimilation; CONV: data assimilation with conventional data only; ATMS:

data assimilation with conventional +ATMS data; CRIS: data assimilation with conventional + CrIS data





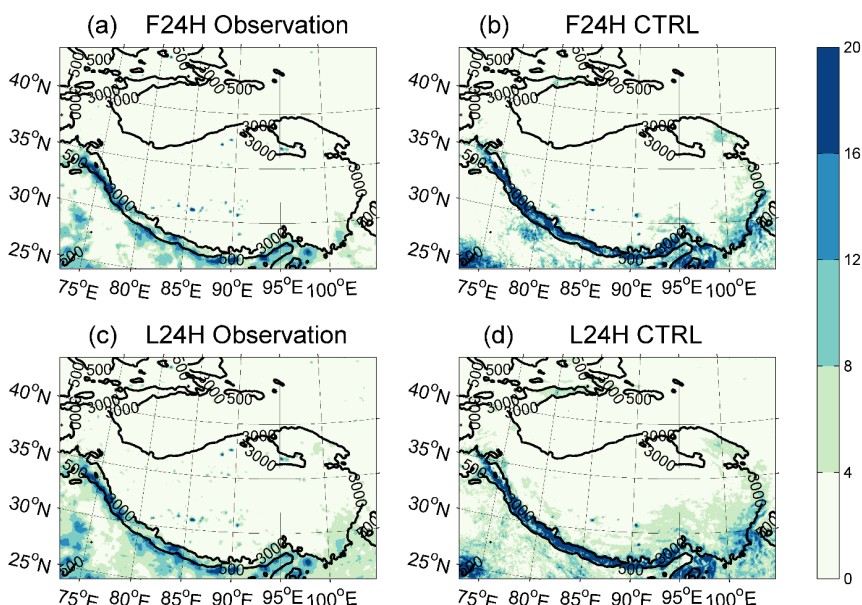


**Figure 4.** Spatial pattern of the monthly mean precipitation (unit: mm) in July 2015. (a), (b) F24H forecast and (c), (d) L24Hforecast. Contours

are altitude (unit: m).





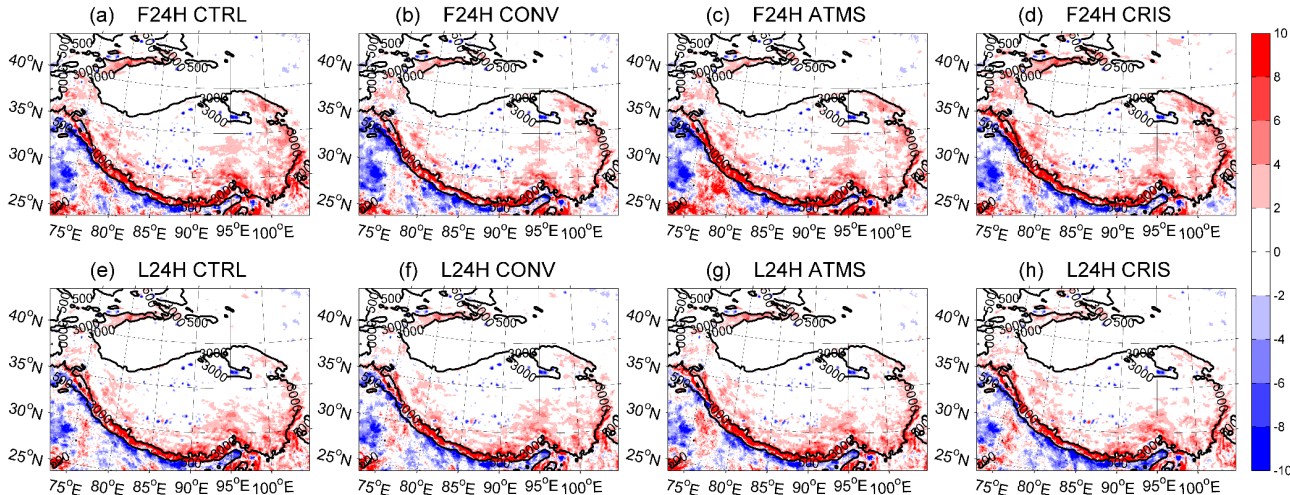

**Figure 5.** Difference value distribution of monthly mean precipitation (unit: mm) during July for data assimilation minus observation experiments.

(a), (e) CTRL minus OBS; (b), (f) CONV minus OBS; (c), (g) ATMS minus OBS for (a)–(d) F24Hforecast and (e)–(h) L24Hforecast. Contours

are altitude (unit: m).





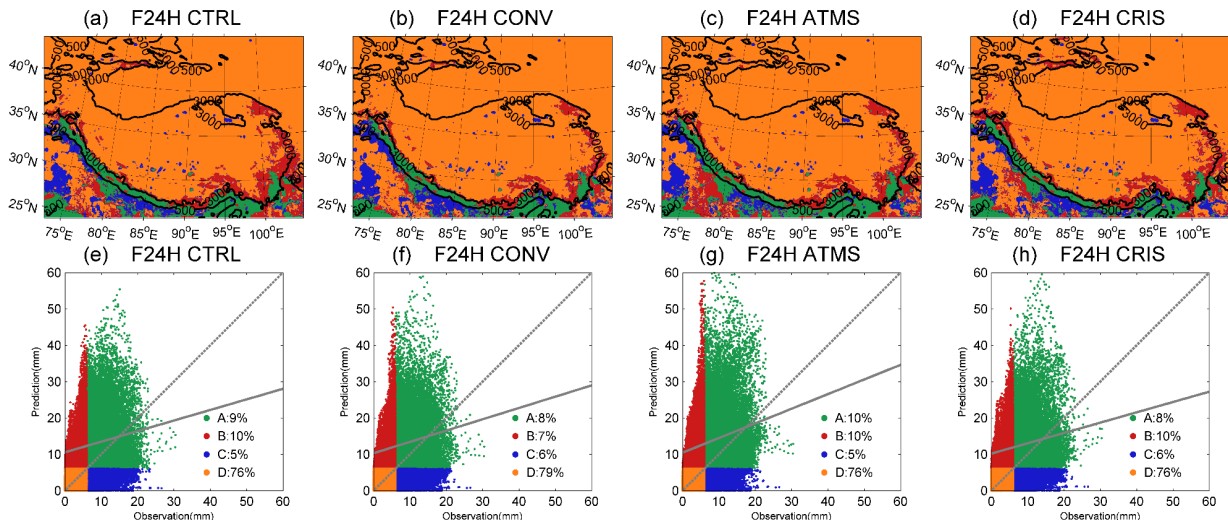

**Figure 6.** Spatial patterns of (a)–(d) the contingency table and (e)–(h) the scatter plots (monthly mean 24 h rainfall over 6 mm threshold is defined

as an "event"). A, B, C and D indicate the Hits, False alarms, Misses and Correct rejections in Table 1, respectively. The solid grey lines indicate

the regression line of A. Contours are altitude (unit: m).





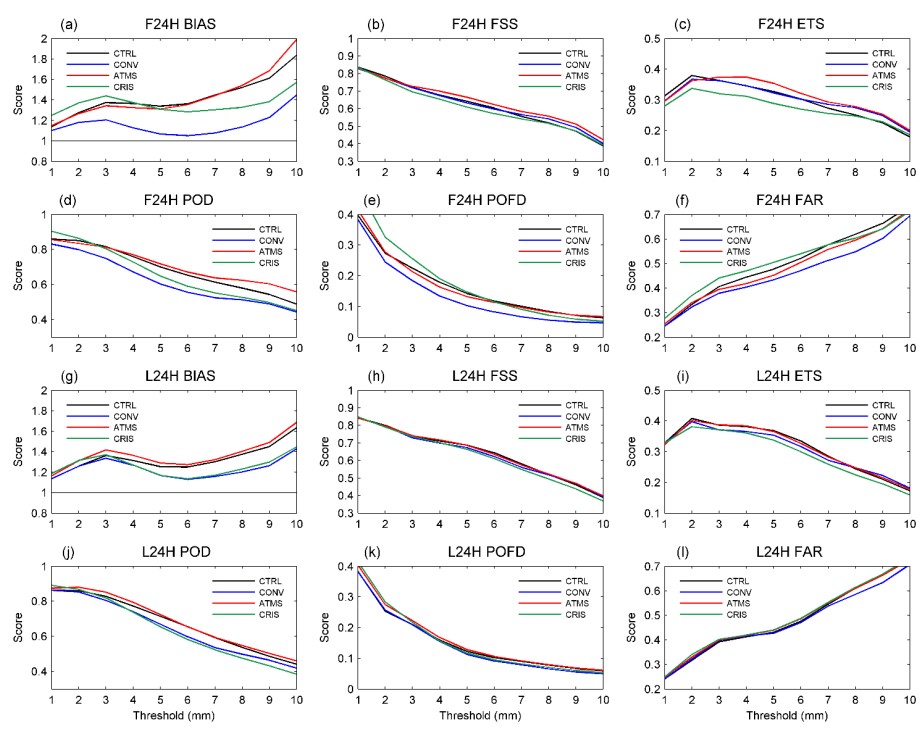


**Figure 7.** Monthly and domain average validation statistics. (a)–(f) F24H forecast and (g)–(l) L24Hforecast.





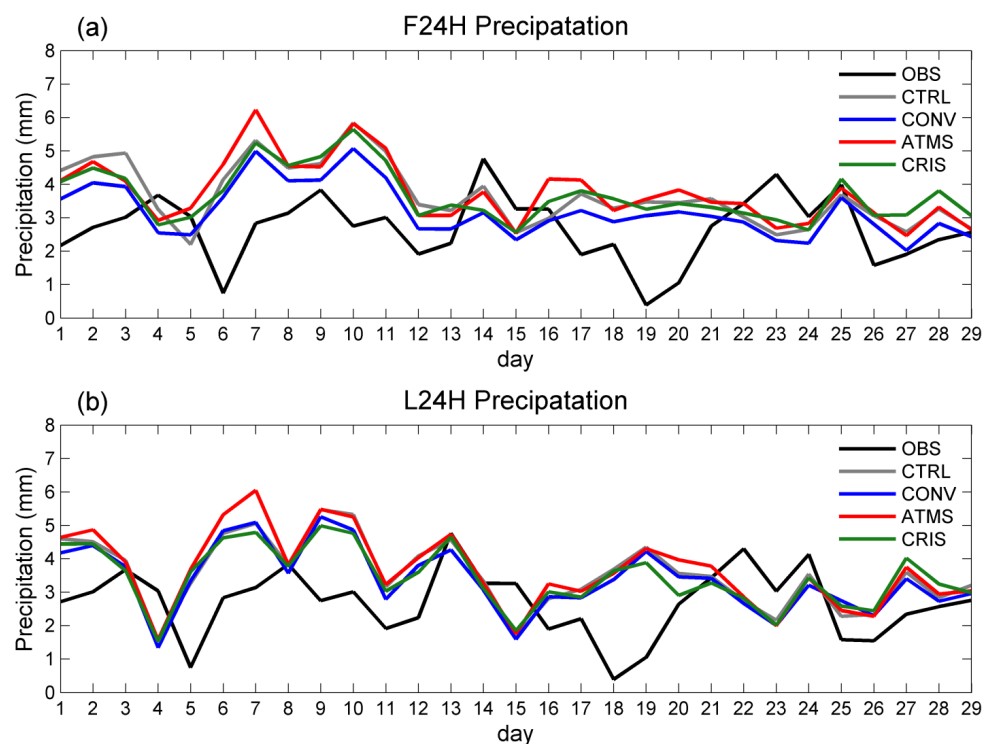


**Figure 8.** Daily precipitation distribution (unit: mm)



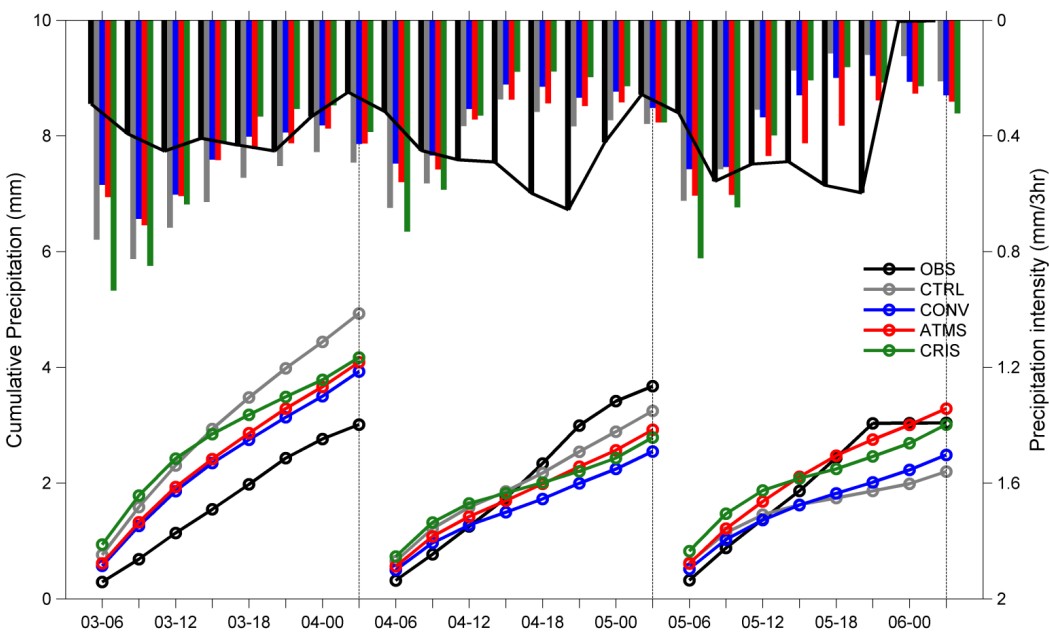


**Figure 9.** Rainfall intensities (bars) calculated for every 3 h amount of precipitation. The cumulative precipitation (curves) is defined as the

precipitation accumulated for each 3 h starting at 06:00 UTC during 3–6 July. The unit is mm.



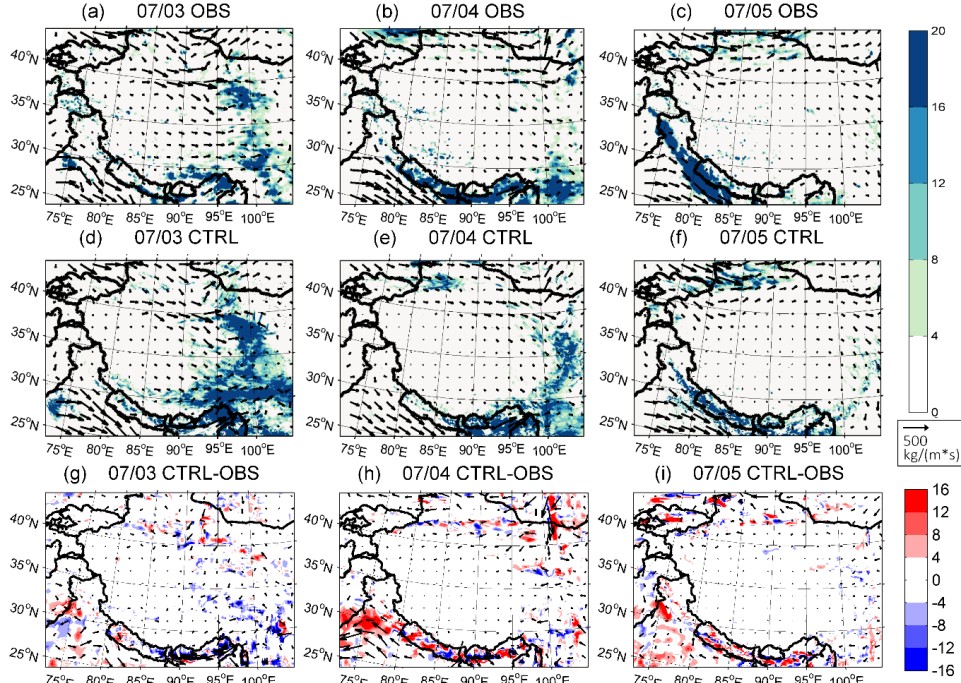

Figure 10. (a)–(f) F24H forecasts of precipitation quantity (shadings) and water vapor flux (vectors) during 3–6 July for (a)–(c) OBS and (d)–(f) CTRL. (g)–(i) Differences in water vapor flux (vectors) and water vapor divergence (shadings) between CTRL and OBS. The unit of precipitation quantity is mm. The units for water vapor flux and divergence is kg/(m*s) and kg/(m2*s), respectively.





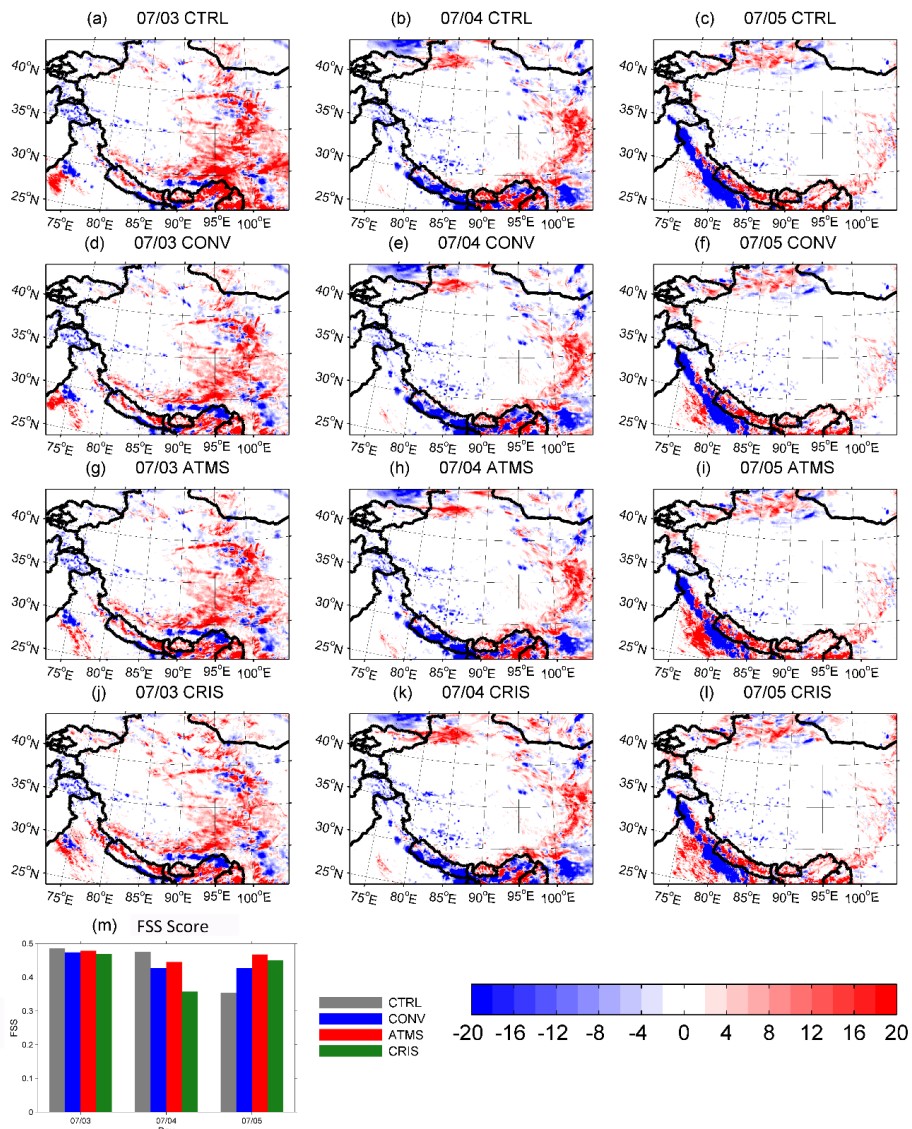

**Figure 11.** (a)–(l) are Differences between  the simulated F24H precipitation and the observed

distribution and (m) is the FSS skill scores with 8 mm threshold during 3–6 July. The unit of

differences is mm.