# Peer review of "An Assessment of the Impact of ATMS and CrIS Data Assimilation on Precipitation Prediction over the Tibetan Plateau"

_Atmospheric Measurement Techniques, 2017_

## Referee Comment (RC1) · Anonymous Referee #1 · 21 Mar 2017

The comment was uploaded in the form of a supplement:
http://www.atmos-meas-tech-discuss.net/amt-2017-31/amt-2017-31-RC1-supplement.pdf

---

## Referee Comment (RC2) · Anonymous Referee #2 · 24 Mar 2017

Review of the paper: An assessment of the impact of AMTS and CrIS Data Assimilation on Precipitation Prediction over the Tibetan Plateau. By: Xue T. et al.

General comment

This paper shows a study on the assimilation of two instruments (advanced technology microwave sounder (ATMS) and cross-track infrared sounder (CrIS)) on precipitation prediction over the Tibetan Plateau (TP) in July 2015. The result shows that AMTS data assimilation improves the results, while the assimilation of CrIS doesn't give better results. The paper is interesting and the arguments well fit the aims of AMT, so it deserves publication on the journal. There are however, some flaws that prevents the publication of the paper in the current form. The main problem is that the argument is

not presented very well, with mistakes and sometimes unclear sentences.

Major points

A thorough review of the English of the paper is needed by a mother tongue. Sentences are sometimes unclear and the language is often not precise. I wrote some errors below (minor points), however I'm sure I missed some of them. The figure caption are often too short and do not explain what it is shown in the figures.

Section 2.1.2 Observation data

Even if there is the reference to a previous study on the performance of the CMORPH dataset for TP (Guo et al., 2014), it is interesting to have some more detail about it, especially about its performance on the TP, considering that the raingauges are sparse over the TP.

Section 4.1 - Lines 255-264

I cannot understand what is shown in Figure 4. If the panels a) and c) are observed values for July they should be the same , while they show different values. Explain.

Section 4.3

Details need to be added on the computations you did in Figure 10, including the mathematical formulation.

Lines 42-45: reformulate the last sentence of the abstract because is not clearly understandable.

Minor points

Line 36: "hourand" -> "hour and".

Line 75: Put a dot after "2008)".

Line 85: two dots after "2013)".

Line 96: change "has" with "had".

Line 113: "The GFS data are ...."

Line 173: two dots after "1998)".

Lines 291-294: The two sentences are unclear. Please, rewrite.

Line 314: put a space between events and "(".

Line 316 and after: I would not call a 6 mm/day precipitation as a "heavy rains". Check thorough the paper.

Line 316: "between" is "among".

Line 359: In the figure 10 the period is 3-5 July and not 3-6. Please change.

Line 374: The score shown is FSS not ETS.

Line 383: Figure 10I does not exist.

Line 385: Change "This phenomenon" with "This result".

Line 386: Figure 10 refers to CNTRL and not to DA experiments, likely you would refer Figure 11.

——— Figures

Figure 2: It is unclear what is shown on the right-y axis. The Figure caption must clearly state what is represented.

Figure 4: The Figure 4 caption must be rewritten. It is unclear. "Spatial pattern of the monthly mean precipitation in July 2015". I believe it is the daily precipitation averaged for the month of July 2015.

Figure 8: title is "precipatation".

Figure 10: the period is 3-5 July not 3-6 July. In the caption, "precipitation quantity" is "precipitation".

---

## Author Comment (AC1) · 7 Apr 2017

Specific comments 1. Lines 108: the content of section 2.1.1 has to be moved into section 2.2.1 of WRF-ARW regional model. Answer: Thanks for your suggestion. We have followed it in the revised manuscript in lines 195-200.

2. Lines 110: Is  $0.5^\circ \times 0.5^\circ$  the finest GFS horizontal resolution available? Answer: Thanks for your question. The NCEP GFS Analysis and Forecast System was upgraded on January 14, 2015 (1200 UTC), providing  $0.25^\circ \times 0.25^\circ$  gridded output which is the finest GFS horizontal resolution available. As the finer GFS product,  $0.5^\circ \times 0.5^\circ$  gridded output is still available. We have made some experiments comparing the winter and summer forecast over Tibetan Plateau while the  $0.25^\circ$  product was not available in

the whole January 2015. So we choose the 0.5 degree gridded output at first.

3. Page.7, line 116: change the title of section 2.1.2 in "Data used for the evaluation/verification" (2.1.2) Answer: We have followed your suggestion in the revised manuscript in line 121.

4. Page.8, line 137: change the title of section 2.1.3 in "Data used for the assimilation" (new 2.1.1) Answer: We have followed your suggestion in the revised manuscript in line 104.

5. Page.10, line 171: change the title of section 2.2.1 in "The GSI 3D-VAR system and the Community Radiative Transfer Model"; please give in this paragraph some theory concepts on GSI 3D-Var system Answer: We have followed your suggestion in the revised manuscript in lines 203-221. Instead of the spectral definition of backgrounds errors in the SSI, GSI is constructed in physical space which the background errors can be represented by a non-homogeneous and anistropic gridpoint and used for both global and regional forecasts. GSI utilizes recursive filters and is designed to be a flexible system that is efficient on available parallel computing platforms (Wu et al., 2002; Purser et al., 2003a,b). The GSI 3D-Var system provides an optimal analysis through two outer iterative minimization of a prescribed function as follows: (1) Where is the analysis state can be calculated by minimizing the penalty function, is the first guess that comes from GFS product in this article representing background model state, are the observations including conventional observation, satellite radiance data, radar data, etc. is the transformation operator from the analysis variable to the form of the error. By means of the two sources of priori data: the first guess and the observations, the solution for the penalty function which indicates the posteriori maximum likelihood estimate of the true atmospheric state can be found. While B and are the error estimates of (covariance matrix of the background error ) and (covariance matrix of the observation error) respectively which are used to weight the analysis fit to individual observations (Wu et al., 2002). Wu, W., R. Purser, and D. Parrish: Three-Dimensional Variational Analysis with Spatially Inhomogeneous Covariances. Mon. Wea. Rev., 130,

AMTD
2905–2916, 2002 Purser, R. J., Wu, W. S., Parrish, D. F., and Roberts, N. M: Numerical aspects of the application of recursive filters to variational statistical analysis. Part I: Spatially homogeneous and isotropic Gaussian covariances. Monthly Weather Review, 131(8), 1524-1535, 2003a Purser, R. J., Wu, W. S., Parrish, D. F., and Roberts, N. M: Numerical aspects of the application of recursive filters to variational statistical analysis. Part II: Spatially inhomogeneous and anisotropic general covariances. Monthly Weather Review, 131(8), 1536-1548, 2003b

6. Line 185: In my opinion this paragraph would follow the one on assimilation data becoming 2.1.3 Answer: We have followed your suggestion in the revised manuscript in lines 155-178.

7. Line 210 section 3.1: please indicate in this section which is the best value for each score Answer: We have followed your suggestion in the revised manuscript in lines 239-262.

8. Lines 242-245: the sentences here are not so clear Answer: Following your suggestion, we have added the text in lines 269-273: The CTRL experiment was carried out first with an initial time of 00:00 UTC and made 54 h forecasts. The data assimilation was applied on the D01 region of the output from CTRL at 06:00 UTC. The initial condition of the DA experiments was derived from the CTRL 6 h forecasts and then DA experiments made a 48 h forecast for each day.

9. Pag.15 lines 275-277: the values in brackets are referred to L24h, is it right? If yes, please specify it in the text Answer: Thanks for your question. The values in brackets are referred to L24h. Following your suggestion, we have added the text in lines 309-312: The overall bias statistic in D02 is 0.97 mm (0.86 mm), 0.52 mm (0.70 mm), 1.08 mm (0.97 mm), and 0.98 mm (0.76 mm) CTRL, CONV, ATMS and CRIS respectively. The values in brackets are referred to L24h.

10. Pag.17 lines 313-316: please indicate on figure 8 (for example using circles or arrows) the overestimated and underestimated events Answer: Thanks for your mention,

AMTD
we have added the grey shadings to indicate the underestimated events in figure 8.

11. Pag. 18 lines 325-326: please add a reference figure Answer: We have followed your suggestion in the revised manuscript in lines 362-367. It is usual to define the amount of 25.0 to 49.9 mm and 50 mm daily precipitation as heavy rain and rainstorm, respectively. However, due to the history data sets of the TP indicating that the days of precipitation exceeding 50 mm are few (only accounting for 0.3% of rain days) (Wei et al., 2003) and referring to previous studies (Wang et al., 2011; Zhao et al., 2015), the heavy rainfall threshold was defined as above 20 mm for the 24 h precipitation in this study. Wei, Z.ïijŇR. H. HuangïijŇW. J. Dong: Interannual and interdecadal variations of air temperature and precipitation over the Tibetan Plateau. Chinese Journal of Atmospheric Sciences, 27(2), 157-170, 2003. Wang, C. H., S. W. Zhou, X. P. Tang, and P. Wu: Temporal and spatial distribution of heavy precipitation over Tibetan Plateau in recent 48 years. Scientia Geographica Sinica, 31(4), 470-477, 2011. Zhao, X. Y., Y. R. Wang, Q. Zhang, and L. Luo: Climatic characteristics of heavy precipitation events during summer half year over the Eastern Tibetan Plateau in recent 50 years. Arid Land Geography, 4, 004, 2015.

12. It would be useful to consider bootstrap confidence intervals when discussing the results Answer: Thanks for these very thoughtful suggestions. To consider bootstrap confidence intervals may be a useful way to present our results. Actually, we calculate these statistics based on a threshold with the different coefficients, please check Figure 7 and the description in lines 327-341 in section 4.1. If we want to consider bootstrap confidence intervals with the different threshold, the calculation should be very complicated. But we accept your suggestion with the different way discussion.

Technical corrections 13. Line 36: a space has to be added between "hours" and "and" Answer: Thanks for pointing out this issue to us. We have corrected it in the revised manuscript in lines 30-31. For the first 24-hour and last 24-hour accumulated daily precipitation.....
14. Line 52: "influences" rather than "influence"; "causes" rather than "cause" Answer: Thanks for pointing out this error to us. We have corrected it in the revised manuscript in line 47. The dramatic modification caused by the rugged terrain influences the local atmospheric circulation and causes strong local convection to arise.....

15ïijŐLine 127: "several" rather than "Several" Answer: Thanks for pointing out this issue to us. We have corrected it in the revised manuscript in line 141. Of the several merged satellite precipitation products.....

16. Lines 138-141-142-143: please specify the acronyms: GDAS, pibal, SSM/I and TCW Answer: Following your suggestion, we have added the text in lines 105-112. The conventional data which is from the Global Data Assimilation System (GDAS)-prepared BUFR files (gdas1.tCCz.prepbufr.nr) is composed of a global set of surface and upper air reports operationally collected by the National Centers for Environmental Prediction (NCEP). It includes radiosondes, surface ship and buoy observations, surface observations over land, pilot balloon (pibal) winds and aircraft reports from the Global Telecommunications System (GTS), profiler and US radar derived winds, Special Sensor Microwave Imager (SSM/I) oceanic winds and atmospheric total column water (TCW) retrievals, and satellite wind data from the National Environmental Satellite Data and Information Service (NESDIS).

17. Line 201: (Table 1) instead of (Table 2) Answer: Thanks for pointing out this mistake for us, we have changed it in the revised manuscript in line 172.

18. Line 280: "(Table 2)" instead of "(Table 1)" Answer: Thanks for pointing out this mistake for us, we have changed it in the revised manuscript in line 236.

19. Line 313: "It can be seen in the time series of Figure 8a that" Answer: Thanks for pointing out this issue for us, we have added it in the revised manuscript in line 350. It can be seen in the time series of Figure 8a that there are four observed heavy rainfall events....
20. Line 317: "The L24H forecasts (fig.8b) showed a...." Answer: Thanks for pointing out this issue for us, we have added it in the revised manuscript in line 355. The L24H forecasts (fig. 8b) showed a similar pattern.

21. Line 322: "the CONV (blue line) experiment" Answer: Thanks for pointing out this issue for us, we have added it in the revised manuscript in line 360. the CONV (blue line) experiment captured the accumulated amount of precipitation....

22. Line 324:"the ATMS (red line) performed the worst....the 24h precipitation maxima" Answer: Thanks for pointing out errors for us, we have added and corrected them in the revised manuscript in line 361 and line 367. the ATMS (red line) performed the worst....the 24 h precipitation maxima surpassing 20 mm are spread in the main precipitation region

23. Line 349: a space has to be added between "experiments" and "data" Answer: Thanks for pointing out this issue for us, we have added a space it in the revised manuscript in line 393.

24. Line 354:"but the results are not the same...." Answer: Thanks for pointing out this issue for us, we have changed the words in the revised manuscript in line 398. but the results are not the same when different data sets are injected.

25. Line 377: "(bottom left in Fig. 11m)" Answer: Thanks for pointing out this issue for us, we have added the words in the revised manuscript in line 437.

26. Line 408: delete the double comma Answer: Thanks for pointing out this issue for us, we have deleted it in the revised manuscript in line 469.

27. Line 409: a space has to be added between "25" and "July" Answer: Thanks for pointing out this issue for us, we have added a space it in the revised manuscript in line 470.

28. Line 445: "we choose" rather than "we chose" Answer: Thanks for pointing out errors for us, we have added and corrected them in the revised manuscript in line 507.
29. Table 1: New Table 1 will be the one about ATMS and CrIS channels and the caption could be modified as follows: "The channels for ATMS and CrIS data that have been selected for the data assimilation procedure" Answer: Following your suggestion, we have revised the text in the caption of new Table 1 in lines 648-649. Table 1. The channels for ATMS and CrIS data that have been selected for the data assimilation procedure

30. Table 2: New Table 2 will be the one about contingency table; please add also more details in the caption Answer: We have followed your suggestion to add more details in the caption of new Table 2 in lines 651-652. Table 2. Rain contingency table used in the verification studies. As a threshold, 6 mm day-1 is chose to separate rain from no-rain events

31. Figure 1: please add more details into the caption of figure 1a (resolution of the domains for example) Answer: Following your suggestion, we have added more details into the caption of figure 1 in lines 661-665. Figure 1. (a) Simulation domains and topography. Resolutions are at 12 km and 4 km for the outer (coarse grid, D01) and inner (nested grid, D02) boxes, respectively. The shading indicates the terrain elevation (unit: m). (b)–(d) Distribution of (b) conventional data observations, (c) scan coverage of ATMS data after data assimilation, and (d) scan coverage of CrIS data after data assimilation at 06:00 UTC on 1 July 2015.

32. Figure 2: clarify into the caption the difference between "observations kept and used" Answer: Thanks for your suggestion, we have revised the caption in lines 666-670 as follows: Figure 2. Blue bars indicate the total amount of radiance read in the DA system. Red bars present the number of kept radiance after first step of quality control. The used percentage after final quality control is shown as black curves. The right y-axis indicates the ratio of used amount to read amount. Top panel is for ATMS (a) and bottom is for CrIS data (b).

33. Figure 3: in the caption there are no info about the part of the figure at the top;

**AMTD**
please also mention into the caption the initial time of each experiment Answer: Thanks for your suggestion, we have revised the caption in lines 671-676 as follows: Top panel shows the schematic of data assimilation configuration with 3D-Var. Bottom panel presents the experiments design. CTRL: control experiment without data assimilation that the initial time is 00:00 UTC from 1 to 31 July; CONV: data assimilation with conventional data only; ATMS: data assimilation with conventional and ATMS data; CRIS: data assimilation with conventional and CrIS data. CONV, ATMS and CRIS experiments all start at 06:00 UTC from 1 to 31 July.

34. Figure 4: put the unit also close to the color bar; "black contours are altitude" Answer: Thanks for your mention, we have put the unit above the color bar now and revised the manuscript in lines 677-678. Figure 4. Daily precipitation averaged (unit: mm) for the month of July 2015. (a), (b) are F24H forecast and (c), (d) are L24H forecast. Black contours are altitude (unit: m).

35. Figure 5: put the unit also close to the color bar Answer: We have put the unit above the color bar now.

36. Figure 7: please list into the caption the validation statistics presented in the figure Answer: The validations statistics is listed into the caption in lines 686-689. Figure 7. Monthly and domain average validation statistics for F24H forecast (a–f) and L24H forecast (g–l). (a) and (g) are Bias Score; (b) and (h) are Fraction skill Score; (c) and (i) are Equitable Threat Score; (d) and (j) are Probability of False Detection; (e) and (k) are Probability of Detection; (f) and (l) are False Alarm ratio.

37. Figure 8: please add more details into the caption Answer: We have added more details in to the caption in lines 690-693: Time series of daily precipitation distribution for F24H forecast (a) and L24H forecast (b). The black, grey, blue, red and green lines indicate observation, CTRL, CONV, ATMS and CRIS, respectively. The unit is mm. The grey shadings indicate the underestimated events.

38. Figure 11: put the unit also close to the color bar Answer: We have put the unit

AMTD
above the color bar now.

39. Can be figures 4-5-6-10-11 a little bit bigger? Answer: Those figures are bigger now.

Please also note the supplement to this comment: http://www.atmos-meas-tech-discuss.net/amt-2017-31/amt-2017-31-AC1supplement.zip AMTD

---

## Author Comment (AC2) · 7 Apr 2017

Major points 1. Section 2.1.2 Observation data: Even if there is the reference to a previous study on the (Guo et al., 2014), it is interesting to have some more details about it, especially about its performance on the TP, considering that the rain gauges are sparse over the TP. Answer: Thanks for these thoughtful suggestions. Due to the gauge distribution is very sparse in TP area, satellite-based estimates have become very important sources for precipitation information. We have further explain the performance of the CMORPH dataset for TP in the revised that paragraph in lines 129-151: Considering the topographically complex terrain characterizing the TP, satellite precipitation data with very high spatial resolution is especially needed. CMORPH product

makes use of the precipitation estimates technique that have been derived from low orbiter satellite microwave observations and geostationary satellite IR data with spatial propagation features. Several studies (Gao et al., 2013; Guo et al., 2014; Tong et al., 2014; Zhang et al., 2015) have compared the CMORPH data with satellite precipitation data sets in the TP area with the conclusion that CMORPH data is one of the most suitable product to use in studying precipitation over the TP. During the period from May to October 2004-2009, Tropical Rainfall Measuring Mission (TRMM) Multisatellite Precipitation Analysis real-time research 3B42 version 6 (TMPA) and CMORPH show better performance in higher correlation and lower RMSE than the Precipitation Estimation from Remotely Sensed Information using Artificial Neural Network (PERSIANN) and its real time version (TMPART) over the TP(Gao et al., 2013). Of the several merged satellite precipitation products (i.e. TMPA, PERSIANN, and the Global Satellite Mapping of Precipitation (GSMaP)), the CMORPH product with the highest resolution (8 km) can capture the afternoon-to-evening precipitation pattern (Guo et al., 2014). Tong (2014) has also compared the performance of four widely-used high resolution satellite precipitation estimates against gauge observations (the CMA data) over the TP during January 2006-December 2012. It's worth noticing that TMPA and CMORPH data had better performance in depicting precipitation timing and amount than the TMPART and PERSIANN at both the plateau and basin scale. Zhang (2015) has also made a conclusion that the high resolution CMORPH data can depict finer regional details, such as a less coherent phase pattern over the TP and better capture the features of the diurnal cycle of summer precipitation compared with TRMM 3B42. Gao, Y. C., & Liu, M. F. Evaluation of high-resolution satellite precipitation products using rain gauge observations over the tibetan plateau. Hydrology & Earth System Sciences Discussions, 2013, 9(8), 9503-9532. Tong, K., Su, F., Yang, D., & Hao, Z. Evaluation ofÂăsatellite precipitation retrievals and their potential utilities in hydrologic modeling over the tibetan plateau. Journal of Hydrology, 2014, 519, 423-437.

2. Section 4.1- Lines 255-264: I can't understand what is shown in Figure 4. If the panels a) and c) are observed values for July they should be the same, while they

show different values. Explain. Answer: Thanks for your attention, we have explained it in the revised manuscript in lines 290-295. Due to the Figure 4 a) standing for the F24H, the first day calculated in Figure 4 a) was during the period of 06:00 UTC 1st July to 06:00 UTC 2nd July and finally ended in the period of 06:00 UTC 29th July to 06:00 UTC 30th July. Therefore the different values in Figure 4 a) and c) can be explained that the Figure 4 c) shows the L24H observed monthly mean accumulated precipitation of which the computing process are different in in two days with Figure 4 a).

3. Section 4.3- Details need to be added on the computations you did in Figure 10, including the mathematical formulation. Answer: Thanks for your suggestion, we have revised the manuscript in lines 403-418 as follows: Zonal component of wind velocity (u), meridional component of wind velocity (v), specific humidity (q), and covariance, which are needed for flux computations, are provided at eight standard pressure levels (1000, 925, 850, 700, 600, 500, 400, and 300 hPa). The equation of unit side length, vertically integrated between the surface level and the top of the atmosphere and averaged in time atmospheric water vapor flux (unit: kg\*m-1\*s-1) can be written as: (11) The zonal and merdional component of vapor flux is described by: (12), and (13), respectively. ÂăWhere ps is the surface pressure and p is the pressure at the "top" of the atmosphere, g is the gravitational constant (9.8 m\*s-2Âă). The water vapor flux divergence (D, unit: kgćm-2ćs-1) is given by: + (14) where a is the radius of the model earth taken as 6371.2 km, Âăis latitude in radians, andÂăÂăis longitude in radians.

Minor points 4. Line 36: "hourand" should be "hour and". Answer: Thanks for pointing out this issue to us. We have corrected it in the revised manuscript in lines 30-31. For the first 24-hour and last 24-hour accumulated daily precipitation.....

5. Lines 42-45: reformulate the last sentence of the abstract because is not clearly understandable. Answer: Thanks for your suggestion. We have revised the abstract in the last sentence into " Overall, based on the experiments in July 2015, the satellite

СЗ

data assimilation improved to some extent the prediction of precipitation pattern over the Tibetan Plateau although the simulation of rainbelt without data assimilation shows the regional shifting."

6. Lines 75: Put a dot after "2008)". Answer: Thanks for your attention. We have revised it in line 71.

7. Line 85: two dots after "2013)". Answer: Thanks for your attention. We have deleted one dot in line 81.

8. Line 96: change "has" with "had". Answer: Thanks for pointing out this mistake for us. We have revised it in line 92.

9. Line 113: "The GFS data are. . ." Answer: Thanks for pointing out this mistake for us. We have revised it in line 199.

10. Lines 291-294: The two sentences are unclear. Please, rewrite. Answer: Thanks for your attention. We have revised it in lines 326-330 as follows: The  $\sim$ 84-89% high percentage of hits and correct rejections events indicates that rainfall events are well predicted. Furthermore, as the false alarms were primarily located in the east of the TP in contrast to the misses in the west, this special pattern can help WRF-ARW model reduce model error in the future which means that WRF-ARW model has promising potential in TP area.

11. Line 314: put a space between events and "(". Answer: Thanks for your attention. We have revised it.

12. Line 316 and after: I would not call a 6 mm/day precipitation as a "heavy rains". Check thorough the paper. Answer: Thanks for your attention. Precipitation is mainly distributed in the south edge of the TP, and the rainfall in other area is very small (Figure 4). The threshold of 6 mm is defined by calculating the whole D02 regional average precipitation so that the value seems relatively small.

13. Line 318: "between" is "among". Answer: Thanks for pointing out this mistake for

us. We have revised it in line 355.

14. Line 359: In the figure 10 the period is 3-5 July and not 3-6. Please change. Answer: Thanks for your attention. We have revised the period in the manuscript in line 403.

1. Line 374: The score shown is FSS not ETS. Answer: Thanks for pointing out this issue for us. We have revised it in line 434.

2. Line 383: Figure 10I does not exist. Answer: Thanks for your attention. We have revised it the manuscript in line 444 as follows: the precipitation experiments all underestimated the amount of precipitation, and CRIS performed particularly badly (Fig. 10c, f, i).

3. Line 385: Change "This phenomenon" with "This result". Answer: We have followed your suggestion in the revised manuscript in line 446 as follows: This result indicates that DA can indeed improve the heavy rainfall forecast.

4. Line 386: Figure 10 refers to CTRL and not to DA experiments, likely you would refer Figure 11. Answer: Thanks for pointing out this mistake for us. We have revised it in line 447: From the above analysis of Figure 9 and 11, it is clear that before the heavy rainfall, DA can improve the simulation of precipitation spatially.

5. Figure 2: It is unclear what is shown on the right-y axis. The Figure caption must clearly state what is represented. Answer: We have followed your suggestion in the revised caption of figure 2 as follows: Figure 2. Blue bars indicate the total amount of radiance read in the DA system. Red bars present the number of kept radiance after first step of quality control. The used percentage after final quality control is shown as black curves. The right y-axis indicates the ratio of used amount to read amount. Top panel is for ATMS (a) and bottom is for CrIS data (b).

6. Figure 4: The Figure 4 caption must be rewritten. It is unclear. "Spatial pattern of the monthly mean precipitation in July 2015". I believe it is the daily precipitation averaged

for the month of July 2015 Answer: We have followed your suggestion in the revised caption of figure 4 as follows: Figure 4. Daily precipitation averaged (unit: mm) for the month of July 2015. (a), (b) are F24H forecast and (c), (d) are L24H forecast. Black contours are altitude (unit: m).

7. Figure 8: title is "precipatation". Answer: Thanks for pointing out this mistake for us. We have corrected the title.

8. Figure 10: the period is 3-5 July not 3-6 July. In the caption, "precipitation quantity" is "precipitation". Answer: Thanks for pointing out this mistake for us. We have corrected the caption. Figure 10. (a)–(f) 24 h forecasts of precipitation quantity (shadings) and water vapor flux (vectors) during 3–5 July for (a)–(c) OBS and (d)–(f) CTRL. (g)–(i) Differences in water vapor flux (vectors) and water vapor divergence (shadings) between CTRL and OBS. The unit of precipitation is mm. The units for water vapor flux and divergence is kg/(m\*s) and kg/(m2\*s), respectively.

Please also note the supplement to this comment: http://www.atmos-meas-tech-discuss.net/amt-2017-31/amt-2017-31-AC2supplement.zip

---

## Referee Comment (RC3) · Anonymous Referee #3 · 12 Apr 2017

Review on "An Assessment of the Impact of ATMS and CrIS Data Assimilation on Precipitation Prediction over the Tibetan Plateau"

General comments:

This paper evaluates the impact of ATMS and CrIS radiance data on the precipitation prediction over the Tibetan Plateau (TP). Since sparse conventional data are available in the TP region, satellite radiance data provide much needed data coverage. With the metrics used in this study, ATMS data are found to have some positive impact in terms of FSS, ETS, and POD, but degrade the system in terms of bias. The use of CrIS data has neutral impact. Overall, this study is interesting and relevant to the scope of AMT. I recommend publication after the following questions are addressed.

Specific comments:

The data usage percentage of CrIS is low. Is the full spectral data file, instead of a subset file, used and read in the GSI?

Section 2.2.1. Model top is set at 10hPa. This may affect the performance of some high peaking channels. Higher model top may be beneficial.

Line 179. The sentence "the ATMS and CrIS satellite radiance data can be read in GSI via CRTM 2.1.3" is not appropriate. After ATMS and CrIS data are read into the GSI, simulated brightness temperatures are calculated via CRTM. The CRTM is considered as observation operator.

Lines 192-195. QC1 is only applied to microwave, a different cloud detection algorithm should also be applied to infrared. Emissivity check is performed not only over ocean but also over land. Regarding QC4, please clarify "retrieved the profiles which meet criterion in QC1 and QC2' – retrieval is conducted? Careful quality control is key to successful data usage.

Figures 4 and 5. The color scheme of the color bars need to be improved. It is not easy to tell different blue/red color levels.

The results in Fig. 5 indicate that, compared to the use of conventional data, the use of ATMS radiance data degrade the monthly mean precipitation, especially in the region of [25N,30N] and [77E, 80E] where conventional data are available. Does this indicate inconsistency between the two types of data? The negative impact of ATMS can also be seen in Fig. 11 (i). The information on the values of ATMS and CrIS observation errors and gross error cut-off will be helpful.

This is a comment – the rainbelt is close to the edges of the D02 domain, not sure if this may affect the results or not.

Due to the forecast model deficiencies, it is shown that it is challenging to improve precipitation forecast. With the water vapor channels available, it would be interesting

to examine their impact on moisture analysis field.

Lines 427-430. Although it is true that microwave can penetrate clouds, I assume only clear-sky radiance data are used in this study.

Reference. Volume and page numbers are missing in one reference. Journal name is also not correct. The correct information is provided below: Zhu, Y., J. Derber, A. Collard, D. Dee, R. Treadon, G. Gayno, J. Jung: Enhanced radiance bias correction in the National Centers for Environmental Prediction's Gridpoint Statistical Interpolation data assimilation system. Quarterly J. Royal. Meteorol. Soc., 140, 1479-1492, 2014.

---

## Author Comment (AC3) · 21 Apr 2017

Referee 3 Specific comments 1. The data usage percentage of CrIS is low. Is the full spectral data file, instead of a subset file, used and read in the GSI? Answer: Yes, you are right, the percentage of the assimilated CrIS data is low. The full spectral data are read in GSI, but the used data is low. In current study, the channels of CrIS are selected according to the NOAA operational system, there are only low percentage channels selected. Then the data will be processed through the data quality control including the data thinning, a large part of data have been kicked out. For example, the clear cloud data will be used, the other part have not been used. So the final used in GSI is very low.

2. Section 2.2.1. Model top is set at 10hPa. This may affect the performance of some high peaking channels. Higher model top may be beneficial. Answer: I totally agree with this point, model top may the performance of some high peaking channels. In our previous experiments, we made the comparison with the different model top (1hPa, 10hPa and 50hPa). We found that the higher model top is used, the more data have been assimilated, but the performance is not really improved, the reason is coming from the regional WRF model limitation. Different from the global model, the regional WRF model don't have a reasonable physical processes at the model top above 10hPa. So the performance with 1hPa model top is quite similar to the 10hPa model top. So we used the 10hPa model top in current study.

3. Line 179. The sentence "the ATMS and CrIS satellite radiance data can be read in GSI via CRTM 2.1.3" is not appropriate. After ATMS and CrIS data are read into the GSI, simulated brightness temperatures are calculated via CRTM. The CRTM is considered as observation operator. Answer: Thanks for pointing out this mistake for us. We have revised the manuscript in lines 226-228 After ATMS and CrIS data are read into the GSI, simulated brightness temperature are calculated via CRTM 2.1.3 in this study.

4. Lines 192-195. QC1 is only applied to microwave, a different cloud detection algorithm should also be applied to infrared. Emissivity check is performed not only over ocean but also over land. Regarding QC4, please clarify "retrieved the profiles which meet criterion in QC1 and QC2' – retrieval is conducted? Careful quality control is key to successful data usage. Answer: The quality control in current study is made according to the GSI used guide. Your comments is right, in GSI, each instrument has its own quality control subroutine. In order to avoid the duplication of the description for the quality control in the user's guide. The original lines 192-195 has been changed in the revised manuscript in lines 163-166. The detailed quality control can be found in the section 8.3 radiance observation quality control in the Gridpoint Statistical Interpolation (GSI) Advanced User's Guide version 3.5 by Developmental Testbed Center

(DTC) (2016). Developmental Testbed Center, 2016: Gridpoint Statistical Interpolation Advanced User's Guide Version 3.5. Available at http://www.dtcenter.org/com-GSI/ users.v3.5/docs/index.php, 119 pp.

5. Figures 4 and 5. The color scheme of the color bars need to be improved. It is not easy to tell different blue/red color levels. Answer: Following your suggestion, we have modified the color scheme in Figures 4 and 5 in the revised manuscript.

6. The results in Fig. 5 indicate that, compared to the use of conventional data, the use of ATMS radiance data degrade the monthly mean precipitation, especially in the region of [25N,30N] and [77E, 80E] where conventional data are available. Does this indicate inconsistency between the two types of data? The negative impact of ATMS can also be seen in Fig. 11 (i). The information on the values of ATMS and CrIS observation errors and gross error cut-off will be helpful. Answer: you are probably right, the result seems that the error in the ATMS experiment is higher than the Conventional data experiment over the specific region. But in general, the precipitation pattern got slightly improved, for example, Figure 6 shows that ATMS looks a little better than the other three experiments but has more extreme-precipitation event forecasts than the others, followed by the CTRL and CRIS, while CONV has the lowest simulation precision. Based on the negative performance in ATMS over the specific region (25-30N, 77-80E), we can't say the two types of data is inconsistency. Because the TP has a complicated terrain pattern, the negative impact may be attributed to data quality control processing, the missing observations or the physical package of WRF-ARW having an inadequate description of snow cover over the plateau surface making the error of margin more prominent (Marteau et al. 2015). In a word, we could not understand the exactly reasons for the negative impact of ATMS according to the specific region. But we can make more experiments examine this problem in the future work. We accept the comment "The information on the values of ATMS and CrIS observation errors and gross error cut-off will be helpful", but in a reality, we hard to understand the value of ATMS or CrIS observation errors in the complicated terrain region, it is a challenge

issue for the next step study.

7. This is a comment – the rainbelt is close to the edges of the D02 domain, not sure if this may affect the results or not. Answer: It is possible that the edges of D02 domain can affect the results, but here the rainbelt is close to the upslope of mountain areas, it is consistent with the observations. This is good comment, we can do some experiments to explore the impact of the location of edges of D02 Domain on the results.

8. Due to the forecast model deficiencies, it is shown that it is challenging to improve precipitation forecast. With the water vapor channels available, it would be interesting to examine their impact on moisture analysis field. Answer: Thanks for your mention. We have analyzed the ATMS and CrIS data assimilation impacts on relative humidity (RH) at 2-meter height above the earth surface in July 2015 in another paper. The results show that the 2-m RH forecasts in July could be modified by assimilating over higher-elevation region which is the part of the TP in this manuscript. This is also a good suggestion, we can make the comparison the water vapor channels with the temperature channels or the other variable channels to examine the impacts.

9. Lines 427-430. Although it is true that microwave can penetrate clouds, I assume only clear-sky radiance data are used in this study. Answer: Yes, you are right. The clear-sky only radiance data is assimilated in this study. But in the processing of the data quality control, the cloud-radiance cannot be completely kicked out, so that the cloud radiance is possible to impact the final results. In our previous study, the result also shows a better performance in microwave radiance experiments compared to the infrared radiance experiments. It is probably related to capability of the clouds penetration for the microwave.

10. Reference Answer: Thank for pointing out this mistake for us. We have revised in the manuscript in lines 645-648.

---

## Referee Report (RR1)

Review of the paper:

**An Assessment of the Impact of ATMS and CrIS Data Assimilation on Precipitation Prediction over the Tibetan Plateau**

By:

Tong Xue, Jianjun Xu, Zhaoyong Guan, Han-Ching Chen, Long S. Chiu, Min Shao

**General comment**

This paper discusses the impact of the DA on the precipitation forecast over the Tibetan Plateau (TP) for July 2015.
In particular, the paper shows the impact of assimilating advanced technology microwave sounder (ATMS) and cross-track infrared sounder (CrIS) satellite data on precipitation prediction for the following two days. The impact of ATMS is positive and apparent, while more controversial results are obtained for CrIS. A justification of this behaviour is given.
The paper was improved compared to the first submission and it is almost ready to be published on AMT. The main problem, in the actual form, is the English, which is sometime not clear and there are sentences that are not clearly understandable. I recommend a general review of the English by a mother tongue.
In the following, there are my specific comments. They are all minor.

**Specific comments:**

Lines 294-295: I would rewrite the sentence as follows to clarify that you are considering the monthly averaged daily precipitation: "It was found that monthly averaged F24H precipitation ranged from 6.0 to 30.4  mm/day, while the monthly averaged L24H precipitation ranged from 6.0 to 29.5 mm/day."

Line 315: …. monthly mean **daily** precipitation ….

Line 322: I would write 6 mm/day to stress better that you are considering a monthly mean of daily precipitation. Also in the Figure 6 caption.

Lines 335-336: I would write: " ….this specific pattern can help improving WRF-ARW forecast in the future."

Lines 348-349: Use "statistics" in place of "methods".

Line 351: …monthly mean **daily** precipitation ….

Line 351: delete "the" before CTRL.

Line 361: use "behaviour" in place of "pattern".

Line 368: the thunderstorm is defined as the precipitation of 50 mm. I guess it is 50 mm or more. Please, check.

Line 378: change "one" with "the".

Lines 412-413: I would write: "The equation of water vapour flux for unit length, integrated from the surface to the top of the atmosphere (kg*m$^{-1}$*s$^{-1}$) is:…"

Line 419: specify which is the "top" of the atmosphere (hPa value).

Lines: 469-471: "Comparisons indicate …". I cannot understand this sentence. Rephrase.

Line 483: "compared with" -> "compared to".

Lines 508-509: "On the other hand…". This sentence is not understandable. Rephrase.

**References**
Line 58: Li et al. 2014 is missing in the references.

Line 69: The reference of Maussion e al. 2011 is incomplete.

Line 76: The reference Eyre et al. 1992 is incomplete.

Line 86: The reference Warner et al. 1997 is missing.

Line 86: Kazumori et al. 2013 is referenced as 2014.

Line 195: There are two papers Zhu et al. (2014) in the references. Check which is the right one.

Line 231: Han, 2006 should be Han et al. 2006.

Line 572: This paper is never referenced in the paper.

Line 574: This paper is never referenced in the paper.

**Figures and captions**

Table 2 caption: "is chosen".

Figure 2: is the ratio expressed as percent on the right y-axis? Clairfy.

Figure 7: In the caption, POD and POFD aren't in the correct order.

Figure 11: In the caption: "… for the 8 mm/day threshold …"

---

## Author Response (AR2)

**General comment**

This paper discusses the impact of the DA on the precipitation forecast over the Tibetan Plateau (TP) for July 2015. In particular, the paper shows the impact of assimilating advanced technology microwave sounder (ATMS) and cross-track infrared sounder (CrIS) satellite data on precipitation prediction for the following two days. The impact of ATMS is positive and apparent, while more controversial results

are obtained for CrIS. A justification of this behaviour is given. The paper was improved compared to the first submission and it is almost ready to be published on AMT. The main problem, in the actual form, is the English, which is sometime not clear and there are sentences that are not clearly understandable. I recommend a general review of the English by a mother tongue.

In the following, there are my specific comments. They are all minor.

Following your recommendation, we have thoroughly revised the manuscript carefully edited its texts with many changes and corrections.

**Specific comments:**

1. Lines 294-295: I would rewrite the sentence as follows to clarify that you are considering the monthly averaged daily precipitation: "It was found that monthly averaged F24H precipitation ranged from 6.0 to 30.4 mm/day, while the monthly averaged L24H precipitation ranged from 6.0 to 29.5 mm/day."

Answer: We have followed your suggestion in the revised manuscript in lines 294-296.

2. Line 315: .... monthly mean daily precipitation ....

Answer: Thanks for pointing out this issue to us. We have corrected it in the revised manuscript in line 316.

3. Line 322: I would write 6 mm/day to stress better that you are considering a monthly mean of daily precipitation. Also in the Figure 6 caption.

Answer: We have followed your suggestion in the revised manuscript in line 322 and Figure 6 caption.

4. Lines 335-336: I would write: " ....this specific pattern can help improving WRF-ARW forecast in the future."

Answer: We have followed your suggestion in the revised manuscript in lines 335-336.

5. Lines 348-349: Use "statistics" in place of "methods". Answer: We have followed your suggestion in the revised manuscript in line 349.

6. Line 351: ...monthly mean **daily** precipitation .... Answer: We have followed your suggestion in the revised manuscript in line 353.

7. Line 351: delete "the" before CTRL.

Answer: We have followed your suggestion in the revised manuscript in line 352.

8. Line 361: use "behaviour" in place of "pattern".

Answer: We have followed your suggestion in the revised manuscript in line 362.

9. Line 368: the thunderstorm is defined as the precipitation of 50 mm. I guess it is 50 mm or more. Please, check.

Answer: Thanks for pointing out this issue to us. We have rewritten it in line 370. It is usual to define the amount of 25.0 to 49.9 mm and superior to 50 mm daily precipitation as heavy rain and thunderstorm

10. Line 378: change "one" with "the". Answer: We have followed your suggestion in the revised manuscript in line 382.

11. Lines 412-413: I would write: "The equation of water vapour flux for unit length, integrated from the surface to the top of the atmosphere (kg\*m-1\*s-1) is:..."

Answer: We have followed your suggestion in the revised manuscript in lines 414-417.

12. Line 419: specify which is the "top" of the atmosphere (hPa value).

Answer: Thanks for pointing out this issue to us. We have rewritten it in line 424. "Where Ps is the surface level and p is the top of atmosphere (10 hPa), g is the gravitational constant (9.8 m\*s-2)."

13. Lines: 469-471: "Comparisons indicate ...". I cannot understand this sentence. Rephrase.

Answer: we change "Comparisons indicate …" into "The pattern, which false alarms are primarily predicted in the east of the TP and the misses in the west, indicates that the WRF-ARW model has promising potential to improve weather forecast ability." in line 479.

14. Line 483: "compared with" -> "compared to". Answer: We have followed your suggestion in the revised manuscript in line 486.

15. Lines 508-509: "On the other hand...". This sentence is not understandable. Rephrase.

Answer: Thanks for pointing out this issue to us. We have rewritten it in lines 511-512.

Moreover, selecting channels is more difficult because of the high altitude, complicated dynamics and thermal conditions.

[revised manuscript text omitted]